# Investigating possibilities for surveillance of long term chlamydia complications in the Netherlands: A qualitative study

Elisabeth Maria den Boogert[1,2]*, Fleur van Aar[1], Janneke C. M. Heijne[1,3,4]

**1** Centre for Infectious Disease Control, Epidemiology and Surveillance, National Institute for Public Health and the Environment (RIVM), Bilthoven, The Netherlands, **2** ECDC Fellowship Programme, Field Epidemiology path (EPIET), European Centre for Disease Prevention and Control (ECDC), Stockholm, Sweden, **3** Department of Infectious Diseases, Public Health Service of Amsterdam, Amsterdam, Netherlands, **4** Amsterdam institute for Immunology & Infectious Diseases (AII) and Amsterdam Public Health research institute (APH), Amsterdam UMC, University of Amsterdam, Amsterdam, Netherlands

* e.den.boogert@ggdhvb.nl

**Data Availability Statement:** All relevant data are within the paper and its Supporting Information file. The original video or audio files and transcripts from the individual interviews cannot be shared

## Abstract

### Objectives

*Chlamydia trachomatis* (chlamydia) is one of the most reported bacterial sexually transmitted infections (STI) worldwide. Chlamydia can cause long term complications such as pelvic inflammatory disease (PID), ectopic pregnancy (EP) and tubal factor infertility (TFI). Changing testing strategies, for example reduced asymptomatic testing, influence chlamydia surveillance, highlighting the need for exploring alternative ways of monitoring chlamydia. We investigated the possibility of introducing routine surveillance of chlamydia related long term complications.

### Methods

A qualitative study including 15 in-depth interviews with a purposive sample of gynaecologists, general practitioners (GP), sexual health and emergency doctors was conducted in the Netherlands in 2021–2022. A semi-structured interview guide focused on experiences with diagnosis and registration of PID, EP and TFI and how a change in asymptomatic chlamydia testing strategy might influence this. Interviews were transcribed and analysed using a thematic approach.

### Results

Analysis showed that gynaecologists most frequently reported diagnosing PID, EP and TFI. Other professions rarely diagnose these complications, with emergency doctors only diagnosing EP. Most respondents reported unique registration codes for PID and EP, but the coding for TFI is more ambiguous. They reflected that diagnosis and registration of PID, EP and TFI are handled differently within their professions. Most respondents acknowledged registration in diagnostic codes as a useful surveillance tool. They expressed concerns in representativeness (e.g. differences in interpretation of diagnosis criteria) and data quality for surveillance.

because the data contains identifying participant information.

**Funding:** The author(s) received no specific funding for this work.

**Competing interests:** The authors have declared that no competing interests exist.

## Conclusions

Patient files of gynaecologists are likely to be most complete for monitoring trends of diagnosed chlamydia related long term complications in the Netherlands. However, when establishing a chlamydia complication surveillance system, professionals should be engaged in further standardizing diagnosis and registration practices. This will improve the quality and interpretability of complication surveillance and facilitate comparison between countries.

## Introduction

*Chlamydia trachomatis* (chlamydia) is one of the most reported bacterial sexually transmitted infections (STI) worldwide [1]. Chlamydia infections can lead to complications such as pelvic inflammatory disease (PID), which subsequently could lead to tubal factor infertility (TFI) or ectopic pregnancy (EP) [2]. Globally, the age-standardized rate of PID prevalence was 53.19 per 100,000 population in 2019 and for EP it was 342.22 per 100,000 population [3]. A study in the United Kingdom estimates that every 1,000 chlamydia infections in women aged 16–44 years, on average, results in 171 extra episodes of PID, 2 more EP and 5.1 more women with TFI at age 44 years [4].

To timely detect chlamydia infections to prevent complications, many countries implemented *chlamydia trachomatis* (chlamydia) control activities ranging from active case finding to screening programmes [5–7]. However, there is an ongoing debate about the impact of these 'test and treat' strategies on reducing chlamydia population prevalence and complications including PID, EP and TFI [7–10]. A possible way to increase knowledge about the impact of control strategies on occurrence of complications is by examining population-level associations between chlamydia control activities and clinical complications [11, 12]. Several countries compared the association between control activities and clinical complications [11, 12]. The authors stress the urgent investment needed to establish surveillance systems to record and monitor trends of chlamydia related complications, such as pelvic inflammatory disease (PID) over time [11]. To our knowledge, no country has implemented routine surveillance of chlamydia related complications.

In the Netherlands, there is a debate about the public health relevance of active case finding by testing for asymptomatic chlamydia infections [7, 10]. This debate is based on lack of practice-based evidence that widespread testing of asymptomatic infected individuals can reduce population prevalence [8, 9]. Also, a number of potential harms from overdiagnosis and overtreatment have been reported, such as psychological stress and potentially unnecessary use of antibiotics, which could lead to antimicrobial resistance [7, 10]. As a result, van Bergen et al. [10] suggest that future strategies should reduce, rather than expand, widespread testing for asymptomatic chlamydia infections. Some countries have already implemented small changes to their testing guidelines, e.g. focussing more on reducing harms from untreated chlamydia infection rather than onward transmission [13]. Australia replaced its focus from testing uptake to strengthening management of diagnosed infections, which involves improving retesting, partner management and PID diagnosis [14].

In the Netherlands, guidelines recommend chlamydia testing for persons experiencing symptoms, their contacts, and risk groups regardless of symptoms such as persons who had unprotected sex [15]. But in the near future, guidelines will recommend to limit asymptomatic chlamydia testing for both men and women. Such a change in chlamydia control activities might impact chlamydia surveillance, which is based on people getting tested for chlamydia at

Sexual Health Centres (SHC) or the general practitioner (GP) [16]. This highlights the need for setting up alternative ways to monitor trends in chlamydia infections. In addition, better insight into chlamydia complication trends is needed when changing the focus from infection control to disease management [10]. Currently, no national surveillance system for PID, EP and/or TFI is in place.

We aimed to investigate the possibilities to introduce long term chlamydia complications (e.g. PID, EP and TFI) under surveillance in the Netherlands according to relevant stakeholders that diagnose these complications. More specific, we aimed to understand the process of diagnosis and registration of these complications and investigate the view of stakeholders on the relevance of a complication surveillance system when changes of chlamydia testing are in place.

## Methods

### Dutch healthcare context

The Dutch healthcare system is based on universal access to care, including compulsory medical insurance for all. All residents of the Netherlands can access care through a basic health insurance package. On top of the mandatory nominal premium, individuals are required to pay a deductible; an amount that insured individuals must pay out of pocket each year before their health insurer begins to reimburse expenses. Some care (e.g. at the General Practitioner) is excluded from co-payment. The General Practitioner (GP) acts as a gatekeeper and can make referrals to medical specialists. Medical specialists (including gynaecologists) are linked to public hospitals [17].

### Design

A qualitative interpretative phenomenological method was used [18]. Phenomenology seeks to understand the individual interpretation and view on a phenomenon [18]. In our study we were interested in individual interpretation of guidelines and registration. Diagnosis of complications is set in guidelines, however interpretation seems difficult [19]. To better understand the experiences of professionals in applying these guidelines and related registrations in their daily practice, we conducted phenomenological interviews.

### Sampling and recruitment

This interpretive phenomenological qualitative study included Dutch medical stakeholders in the diagnosis and registration of chlamydia related complications identified by a stakeholders analysis conducted at the start of the study. In the stakeholder analysis the researchers identified relevant stakeholders and classified their potential influence and interest towards the subject. Stakeholders with low and high influence combined with high interest were invited. This resulted in inviting gynaecologists, GPs, doctors working at a regional Sexual Health Centre (SHC) and emergency room (ER) doctors to participate in the study through network and snowball sampling. The sample size was determined by the principle of saturation for the main themes of the interview guide (see S1 File for interview guide). When saturation was not reached and recruitment was difficult, medical doctors working under supervision of a gynaecologist, GP, ER doctor or SHC doctor were recruited.

### Data collection

Data were collected using a semi-structured interview guide for individual face-to-face or online interviews was developed. Participants received a leaflet that explained the study and

their privacy rights. After verbal consent for recording, the interview was either videotaped or audiotaped. After starting the recording, respondents expressed their consent for participating in the study. The interview guide consisted of four main items (S1 File);

1. diagnosis process (e.g. related symptoms, standard STI test), with the main theme: does complication diagnosis change when chlamydia testing strategies change?

2. registration process (e.g. registration codes, standardized practice), with the main theme: is registration reliable and uniform within and across professions?

3. importance of complication surveillance (e.g. knowledge on trends, use of surveillance information), with the main theme: do professionals acknowledge the need for complications surveillance?

4. possibilities for complication surveillance (e.g. implications of data sources, obtaining necessary information for surveillance), with the main theme: getting a first insight into needs and possibilities regarding complication surveillance

New relevant topics raised by the interviewees in the first interviews were added to the interview guide. Profession, region of workplace (province-level), gender and age group were registered. The interviews lasted between 20 and 75 minutes. Respondents were recruited and interviews were conducted from 9 December 2021 to 20 July 2022 by the first author. The research team consists of three epidemiologists working in infectious diseases and STI research. The interviewer had no prior interaction with the respondents regarding the themes described, however some respondents were part of the network of the research team, the interviewer and other respondents.

### Analysis

All interviews were transcribed by a professional transcription firm. Transcripts of recordings were checked by the first author to ensure accuracy. To identify individual experience with interpreting guidelines and registrations in our qualitative dataset we conducted an interpretive thematic analysis using MAXQDA Plus 2022 (Release 22.0.0) for data management [20]. Two coders generated codes based on the interview guide and double-coded one transcript. After discussion, a new code book was generated and tested by double-coding a different transcript. After agreement of the code book, one coder continued to code all other transcripts and discussed uncertainties with the second coder, and, if there was no consensus, a third person was consulted.

### Ethical approval

This study was reviewed by the Medical Ethical Committee Utrecht, the Netherlands, which judged that it was not subject to the law for Medical Research Involving Human Subjects (21-705/C). All respondents provided informed consent for participation.

### Results

Fifteen in-depth interviews with a purposive sample of gynaecologists (n = 3), medical professionals under supervision of a gynaecologist (n = 2: in vitro fertilization (IVF)-doctor and fertility doctor), GPs (n = 6), doctors at a regional SHC (n = 3) and emergency room (ER) doctor (n = 1) were conducted. Gynaecologists, IVF and fertility doctors were grouped and referred to as gynaecologists (n = 5). Ten respondents were female (67%), and ten respondents were 20–39 years old, five were 40 years or older. Respondents worked in seven of twelve provinces in the Netherlands.

## Pelvic inflammatory disease (PID)

**Diagnosis.** According to all respondents, patients with symptoms related to PID most often consult a GP and/or gynaecologist, sometimes via the emergency room. All GPs see patients with suspected PID, and all but one GP would always refer a patient to the gynaecologist for further diagnosis and treatment (Table 1). This GP reported occasionally diagnosing and treating PID without interference of a gynaecologist. All respondents acknowledge the difficulty of diagnosing PID, as symptoms are often non-specific and there is no easy, non-invasive diagnostic test. "it is not as if there is an easy test that tells you it is a PID" (Gynaecologist).

All GPs were aware of the guideline on diagnosing PID. Some GPs strictly adhered to the guideline with respect to PID symptoms, and expect other GPs to diagnose accordingly. "I hardly ever diagnose or register a PID, because I, as I said, almost never see those classical three; stomach ache, fever and an STI" (General practitioner). Other GPs were less conclusive on what specific symptoms to include for a final diagnosis, and talked more about the overall feeling and the patient's story. Most respondents suspect a PID based on symptoms and anamnesis (i.e. GPs and gynaecologists) and, if PID is suspected, perform a chlamydia PCR test to identify a current chlamydia infection. GPs were not unanimous on always performing a chlamydia PCR test, one GP assumed that gynaecologists test for a chlamydia infection after the GP referred the patient to them and another GP said they will only test if the patient has a high STI risk. All SHC doctors reported seeing an occasional PID and referring this patient to the gynaecologist for further diagnosis and treatment.

## Registration

All GPs use International Classification of Primary Care (ICPC) codes to register the patient's diagnosis, but they use different versions of software to register the ICPC codes. PID has a specific ICPC code, that can be administered to the related episode in the patient's file. Gynaecologists, SHC doctors and the ER doctor think that the GP has the most complete patient files related to complications since they get feedback letters from other professionals to add to their patient's file. However, most GPs were unsure if the code is administered correctly. Usually a patient comes to the GP with lower abdomen pain, and is later diagnosed, mostly by the gynaecologist, as PID. All GPs and gynaecologists confirm that gynaecologists always send a feedback letter to the GP including the confirmed diagnosis. GPs should then, based on the information in this letter, change the registration code for this patient, from abdomen pain to PID. All GPs, except one, were unsure if changing the registration code was strictly done by their colleagues, but most said they do change the code themselves.

All gynaecologists report there is a specific Diagnostic Treatment Combination (DBC) code for PID, although they do not know the code by heart. Three gynaecologists said that, besides the DBC coding, they also use the National Specialistic Fertility Records (LSFD), which has more specific and detailed codes than the DBC coding.

## Ectopic pregnancy (EP)

**Diagnosis.** GPs reported never diagnosing an EP and always referring a suspected EP to the emergency room (Table 1). The ER doctor confirms diagnosing and treating acute EP, with follow-up by the gynaecologist. All gynaecologists reported seeing EP, mostly as part of a fertility treatment and in the emergency room. SHC doctors reported never seeing or diagnosing EP (Table 1). A positive pregnancy test is always part of the diagnosis process; performing a Chlamydia Antibody Test (CAT) or chlamydia PCR is not always indicated. "I expect the gynaecologist to do that [perform an STI test or consult on STI history]" (General Practitioner).

**Table 1. Overview of diagnosis and referral of complications according to medical professionals (n = 15).**

|  | Pelvic inflammatory disease | Ectopic pregnancy | Tubal factor infertility |
|---|---|---|---|
| **Diagnosis of complication** |  |  |  |
| Sexual health centre doctor | Rarely | No | No |
| General practitioner | Yes | No | No |
| Gynaecologist | Yes | Yes | Yes |
| Emergency room doctor | No | Yes | No |
| **Referral of patients with complications** |  |  |  |
| Sexual health centre doctor | to gynaecologist | - | - |
| General practitioner | to gynaecologist | to emergency room and/or gynaecologist | to gynaecologist |
| Gynaecologist | - | - | - |
| Emergency room doctor | - | to gynaecologist | - |

**Registration.** In the DBC coding system, EP is a clustered code with abortion, EP, hyperemesis gravidarum and other pathology related to the first 16 weeks of pregnancy. The LSFD system used by some gynaecologists and the ICPC system used by GPs, both have a specific EP code. The ER doctor uses DBC codes and registers the treatment and referral of the EP to the gynaecologist in the patient's file. The gynaecologists that use both LSFD and DBC register in both, with LSFD being more specific for EP.

## Tubal factor infertility (TFI)

**Diagnosis.** According to the respondents, when patients have been trying to become pregnant for over a year, patients consult the GP and the GP refers them to the gynaecologist. SHC and ER doctors report that they do not diagnose TFI in their patients (Table 1). According to most; follow-up diagnostic tests at the gynaecologist involves a Chlamydia Antibody Test (CAT), however, one gynaecologist said they would do a PCR instead. If this test is positive, and the cause of infertility is not related to the male partner, female patients can opt for a transvaginal hydro laparoscopy (THL) or a hysterosalpingography (HSG) for further diagnosis depending on the hospital. Gynaecologists mentioned that these diagnostic tests can also be offered as a treatment option as they supposedly can open up the Fallopian tube. Some hospitals offer this test without a positive CAT.

**Registration.** Gynaecologists report that the registration of TFI is ambiguous. The DBC coding system has more general codes, such as initial fertility testing, with a subcode called subfertility. According to one gynaecologist this subcode is not registered properly; "Subfertility caused by tubal pathology is a specific DBC code, but is registered very poorly". The LSFD has a more specific coding system, in which you can register the cause of subfertility as tubal factor infertility. The GP registers a referred patient under subfertility. Diagnostic tests, such as HSG and THL are registered in the patient's file, however more general as diagnostic test and not with a specific code stating HSG or THL. In the LSFD these tests are registered as specific codes.

**Surveillance.** Most respondents acknowledge the importance of complication surveillance to keep track of possible changes in trends in chlamydia related complications and named many possible methods and sources for obtaining information, such as the data collection at the health insurers, obtaining data based on the ICPC or DBC codes from the medical practice (hospital and/or GP) or setting up cohort studies.

Most understand the benefit of surveillance on long term complications, however would not immediately see it contributing to their daily work. When thinking about it more, most said they would use it to inform their patients about their risk of getting or having a

complication. However, most respondents acknowledge that it is hard to determine the causality between a (previous) chlamydia infection and a complication.

When discussing what would change in diagnosing complications and in trends of complications if the chlamydia control strategy would change to less asymptomatic testing, all but one respondent were clear that the complication diagnosing process would not change, as symptoms would still occur. However, some expressed concerns that for example less testing, diagnosing and treating of chlamydia could result in more undiagnosed chlamydia and consequently more complications. One GP expected that less chlamydia testing and diagnosing could lead to missed or delayed diagnoses of complications, as the GP would not be triggered to think of a complication when a previous chlamydia infection is not reported in the patient's file. One gynaecologist expressed concerns about increasing chlamydia prevalence and subsequently an increasing number of positive Chlamydia Antibody Tests (CAT). This would result in an increase of invasive fertility testing as, according to protocol, a positive CAT is a proxy for a HSG or THL diagnostic test. According to this gynaecologist, this may lead to an increase of false positive HSG results, and an increase in, possibly unnecessary, laparoscopies.

## Discussion

According to our respondents, gynaecologists diagnose PID, EP and TFI most frequently compared to other health professionals. Other professions rarely diagnose these complications and mostly refer patients with a suspected complication to the gynaecologist. Emergency doctors only diagnose EP. Most respondents reported unique registration codes for PID and EP, but the coding for TFI is more ambiguous. These are mostly general complication codes, not necessarily related to a (previous) chlamydia infection. Respondents reflected that diagnosis and registration of PID, EP and TFI are handled differently within their profession. Most respondents acknowledged registration in diagnostic codes as a useful surveillance tool.

Our study is the first to explore possibilities for setting up surveillance of PID, EP and TFI by including multiple relevant stakeholders from different regions of the Netherlands. We included both newly started and more experienced professionals with broad coverage in the country. A possible limitation of this study is that doctors who responded were more interested in the theme and might be more knowledgeable about the topics. We do not believe that this influences the results about diagnosing or registration, but they might voice a more positive tone towards setting up a surveillance system. Second, a possible limitation is the sample size. It was difficult to include gynaecologists as they were often occupied with clinical work and were not able to clear their schedule. For this reason, we included professionals in training or under supervision of a gynaecologist. We believe, however, that they have enough experience to adequately answer our questions, as their answers were in line with the three gynaecologists that were included. Based on the interview with the ER doctor, the interviews with other stakeholders, and informal communication, we decided not to include more ER doctors. We concluded that they were not relevant stakeholders in the diagnosis and registration of complications in the Netherlands. Saturation for the main themes was reached after two interviews with SHC doctors, mainly since they explained that they hardly ever diagnose chlamydia related complications. A third interview was conducted to confirm their responses regarding the main themes. However, since network and snowball sampling could result in a more limited specificity, and thus requiring more participants to reach saturation, it might still be possible that other views among SHC doctors exist [21]. Furthermore, we believed that saturation for GPs and gynaecologists was reached for the main themes. For example, professionals were unanimous that diagnosis of complications would not change if chlamydia testing strategies would change (main theme 1). For other topics, such as whether or not all GPs would refer a

suspected PID to the gynaecologist, a larger sample size could yield a wider range of views than reported in this article.

Currently two potential existing systems based on registration codes could be implemented for surveillance of complications purposes. First, a sentinel surveillance system, called Nivel (the Netherlands Institute for Health Services Research) sentinel stations, exists in a small proportion (around 8%) of GP offices in the Netherlands with a patient population registered at these practices being representative for the Dutch population in terms of age and gender. This source includes routinely extracted data from electronic health records (consultations and morbidity in ICPC codes) and is already in use to estimate the number of chlamydia diagnoses performed at GPs. Second, for data of patient's medical records from hospital registries, the Dutch Hospital Data (DHD) can be contacted for a data request for research or surveillance purposes. The DHD uses ICD-10 coding for medical data (e.g. diagnosis and treatment) and has nationwide coverage. All gynaecologists in the Netherlands are linked to a hospital and thus their patient's medical records are present in the DHD. In our study all hospital staff talked about DBC coding as the standard practice for registering diagnoses. DBC coding is a clustering of ICD-10 codes with an automated connection to the ICD-10 codes used in the DHD. ICD-10 and DBC coding are used by all hospitals in the Netherlands as the standard registration practice for medical records and are a Dutch coding system used for billing healthcare services provided to the patient.

These hospital and GP data sources could be used to implement a surveillance system on long term complications of chlamydia. However, professionals identified potential concerns regarding data quality, as codes for EP and TFI can include other diagnoses and not all PID is registered correctly by the physician in the correct registration codes (e.g. in patient files at GP offices). Furthermore, we also identified concerns regarding representativeness of the registration system, as we observed differences in interpretation of diagnostic criteria within and between professions. For example, one GP was very clear about only diagnosing and registering a PID if a patient presented the three classical criteria (i.e. stomach ache, fever and an STI), whereas another GP would diagnose a PID based on the overall feeling and the patient's story.

When considering the different perspectives and professions in the diagnosis of complications, we see that most complications are diagnosed by the gynaecologists. Gynaecologists, SHC doctors and the ER doctor think that the GP has the most complete patient files related to complications as they will get feedback letters from other professionals to add to their patient's file. However, GPs did not feel confident that their patient files are most complete related to PID, EP and TFI compared to the other professions. GPs stated that the registration code for the complication might not always be administered or changed in the patient files after receiving the feedback letter from the gynaecologist. Based on what respondents told us, we feel that gynaecologists, more than GPs, actually diagnose the complications, and properly register them in their hospital databases. Furthermore, as GPs hardly ever diagnose chlamydia complications themselves, but rather irregularly register the diagnosis retrospectively when receiving a feedback letter from the gynaecologist, we expect great overlap between the diagnoses in GP and DHD registration systems. We therefore expect the patient files of gynaecologists to be most complete for monitoring trends of diagnosed chlamydia related long term complications and thus the DHD registration most suitable for complication surveillance.

When considering complication surveillance, an important aspect should be taken into account. Possible observed changes in PID, EP and/or TFI trends cannot be directly linked to changes in incidence of chlamydia infections, as there could be many other explanations for the observed trends. Especially since other infections can also lead to PID, EP and TFI, such as gonorrhoea [22]. Other explanations could include changes in access to health care, (registration of) diagnosis, or changes in treatment of chlamydia infections [23]. Therefore, one should

be careful in interpreting trends of PID, EP and TFI when complication surveillance is in place.

## Conclusion

Patient files of gynaecologists are likely to be most complete for monitoring trends of diagnosed chlamydia related long term complications in the Netherlands. However, when establishing a chlamydia complication surveillance system, professionals should be engaged in further standardizing diagnosis and registration practices. This will improve the quality and interpretability of complication surveillance and facilitate comparison between countries.

## Supporting information

**S1 File. Semi-structured interview guide–surveillance of chlamydia related long term complications.**
(PDF)

## Acknowledgments

The authors would like to thank all respondents for their interesting and insightful thoughts to help our research.

## Author Contributions

**Conceptualization:** Elisabeth Maria den Boogert, Fleur van Aar, Janneke C. M. Heijne.

**Data curation:** Elisabeth Maria den Boogert, Fleur van Aar.

**Formal analysis:** Elisabeth Maria den Boogert.

**Investigation:** Elisabeth Maria den Boogert.

**Methodology:** Elisabeth Maria den Boogert, Fleur van Aar, Janneke C. M. Heijne.

**Project administration:** Elisabeth Maria den Boogert.

**Supervision:** Fleur van Aar, Janneke C. M. Heijne.

**Validation:** Elisabeth Maria den Boogert.

**Writing – original draft:** Elisabeth Maria den Boogert.

**Writing – review & editing:** Elisabeth Maria den Boogert, Fleur van Aar, Janneke C. M. Heijne.

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
