## [Decision Letter · Decision Letter 0]

4 Feb 2024

PONE-D-23-23499Investigating opportunities for surveillance of long term chlamydia complications in the Netherlands: a qualitative studyPLOS ONE

Dear Dr. den Boogert,

Thank you for submitting your manuscript to PLOS ONE. After careful consideration, we feel that it has merit but does not fully meet PLOS ONE’s publication criteria as it currently stands. Therefore, we invite you to submit a revised version of the manuscript that addresses the points raised during the review process.

We look forward to receiving your revised manuscript.

Kind regards,

Jesse L Clark, MD, MSc

Academic Editor

PLOS ONE

Journal Requirements:

Additional Editor Comments:

Specific areas to address in your revision include expansion of the Background/Introduction to better situate your manuscript within the existing healthcare context, adding detail to the Methods/Results to expand on data collection and analysis (particularly with regard to data saturation), and correction of all typos/grammatical errors in the manuscript.

Reviewers' comments:

Reviewer's Responses to Questions

**Comments to the Author**

1. Is the manuscript technically sound, and do the data support the conclusions?

Reviewer #1: Partly

Reviewer #2: Yes

2. Has the statistical analysis been performed appropriately and rigorously? 

Reviewer #1: N/A

Reviewer #2: N/A

3. Have the authors made all data underlying the findings in their manuscript fully available?

Reviewer #1: No

Reviewer #2: Yes

4. Is the manuscript presented in an intelligible fashion and written in standard English?

Reviewer #1: No

Reviewer #2: Yes

5. Review Comments to the Author

Reviewer #1: General comments:

Thank you for the opportunity to review this interesting manuscript on opportunities for surveillance of long term chlamydia complications in the Netherlands. The manuscript presents findings of a qualitative study among 15 healthcare professionals to explore their views on potential future routine surveillance of chlamydia related long-term complications, including pelvic inflammatory disease (PID), ectopic pregnancy (EP) and tubal factor infertility (TFI).

Given current debates on a paradigm shift from Chlamydia infection control to disease control, I believe that this is a timely topic of great public health relevance and general interest to PLOS ONE readers. However, I think that for the more general reader, it would be good if the authors could add further information to the introduction/discussion sections, including regarding limitations on the use of complications surveillance to compensate for reduced screening of asymptomatic infections. It is also important to acknowledge methodological limitations and ensure that relevant guidance on reporting of qualitative studies has been followed.

Specific comments:

1. Objectives in abstract and introduction:

I wonder whether the objectives could be stated a bit more clearly (possibly seeking support from a native speaker for this – well aware that I am not a native speaker, myself). I am not sure, for example, whether the word ‘opportunity’ is the best word to capture/specify what the authors mean – were you thinking of the word more in the sense of possibility/ options or in the sense of a favourable chance or circumstance and prospect for success? In the abstract, the objective ends with ‘aiming to evaluate the impact of changing chlamydia testing strategies on occurrence of complications.’ I think this refers to the routine surveillance of complications (i.e. complications surveillance could be used to evaluate the impact of potential future changes in testing strategies), but the quick reader might think that your qualitative study aims to evaluate to impact of changing testing strategies…

2. Introduction/ Discussion:

a. You could consider briefly mentioning the prevalence and trends of Chlamydia complications globally and/or nationally (e.g. possibly He et al 2023 for the global burden of PID and EP, doi: 10.1186/s12889-023-16663-y)

b. The general reader might be interested to know whether any other countries have already changed testing strategies and/or have already some form of Chlamydia complications surveillance in place and what their experiences is with this?

c. You could consider discussing surveillance of Chlamydia complications with regards to certain limitations, including relating to the correlation of Chlamydia infections with PID and with subsequent EP and/or TFI. For example, trends could be discordant due to changes in the health care system other than STI testing, e.g. regarding delayed or inadequate treatment of Chlamydia infection and/or PID. (For changes in access to hospital care, see for example the paper by Chen et al 2005 doi: 10.1136/sti.2004.012807 ). In the case of TFI, possibly changing pregnancy rates would also need to be considered when analysing data.

d. It would also be good if the more general reader could have an idea on how likely it is that it can be differentiated that PID, EP and/or TFI are due to Chlamydia, and not due to other infections, such as Gonorrhoea (also keeping gonorrhoea-related antimicrobial drug resistance in mind), or bacterial vaginosis-associated bacteria, or other causes, such as endometrioses (keeping changing trends in endometriosis rates in mind)? Having said all this, I believe that PID, EP and TFI changes are worth monitoring, regardless of its cause, but issues around cost-effectiveness might need to be discussed.

e. Line 264-265 reads “Based on what respondents told us, we feel that gynaecologists, more than GPs, actually diagnose the complications, and properly register them in their hospital databases”. International readers might not be sure why gynaecologists would be able to register complications in ‘their hospital database’, as they might not know whether in the Netherlands gynaecologists are usually based at hospitals (including for outpatient care) or in (private or public) clinics/ practices separate from hospitals. It would therefore be good to briefly provide in the introduction or discussion some background information on the Dutch health care system, also in terms of access to free health care.

f. Generally, I think the manuscript is well written, I have only spotted several typos/ grammar/spelling errors, and it is necessary to cross-check the whole document, e.g. in line 135, 143,147 and 256 etc need to say “patient’s” or “patients’” instead of “patients”, line 113 reads “in-dept interviews” instead of “in-depth interviews”, line 188 should say “tubal pathology” instead of “tuba pathology”, and in line 165 please clarify what is meant by “red.” In “red. perform an STI test…”. In line 83, please indicate the meaning of the acronym SHC at first use, even if most readers will guess what is meant. In line 240 it is not clear what ‘NIVEL’ means.

3. Methods:

a. Following reporting guidance: Guidance by different journals varies, but I think it would be good if you could indicate at the beginning of the methods section that you have used relevant international guidelines/checklists for the reporting of qualitative research. Therefore, if the editor agrees, please upload with your rebuttal letter a SRQR checklist(http://journals.lww.com/academicmedicine/fulltext/2014/09000/Standards_for_Reporting_Qualitative_Research___A.21.aspx) or the COREQ checklist (https://academic.oup.com/intqhc/article/19/6/349/1791966/Consolidated-criteria-for-reporting-qualitative), indicating the page/line numbers of your manuscript where the relevant information can be found, updating the manuscript as needed to ensure all reporting requirements are met.

For example, I noted that there is insufficient information on your qualitative approach/ research paradigm and researcher characteristics and reflexivity.

b. Sample size and saturation: Line 85 indicates that the sample size was determined by the principle of saturation and that where saturation was not reached or was difficult, medical doctors working under supervision of a gynaecologist were recruited. Overall, I think that the sample sizes for the different specialities in different regions of the country are rather low and it seems unlikely that saturation (however defined) has been reached for all relevant concepts (although it might have been met for some of the most important themes, but this would need to be more clearly stated). Generally, it seemed that the reported results are quite descriptive and do not go much into depths. Nevertheless, I think you have retrieved important information from this research, it may only need further information and better acknowledgement of its limitations. For example, you found that one of the six GPs said they sometimes diagnose and treat PID without referral to a gynaecologist, which would mean that the diagnosis would not be captured by gynaecologists/ in a hospital database. It is unclear whether in a bigger sample, more GPs would have reported similar behaviour.

The concept of ‘data saturation’ is well-debated in qualitative methodology literature and it would therefore be good if you could define this term more clearly and your decisions around it in your paper. Did you have an initial sample size you were aiming for? How did you decide (before and/or during data collection) your definition of data saturation to conclude recruitment, and how did you measure this? How could you be sure there were no more relevant themes to come, or that further interviews could not contribute to refining of the themes? Please see for example the references below regarding the current debate on ‘saturation’ and sample sizes in qualitative research:

Saunders B, et al. Saturation in qualitative research: exploring its conceptualization and operationalization. Qual Quant. 2018;52(4):1893-1907. Epub 2017 Sep 14. DOI: 10.1007/s11135-017-0574-8

Tight, M. (2023). Saturation: An Overworked and Misunderstood Concept? Qualitative Inquiry, 0(0). https://doi.org/10.1177/10778004231183948

Malterud K, et al: Sample Size in Qualitative Interview Studies: Guided by Information Power. Qual Health Res 2016, 26(13):1753-1760. DOI: 10.1177/1049732315617444

Reviewer #2: Thank you for the opportunity to review this manuscript which uses qualitative methods to examine clinician practices and behaviors related to diagnosis and reporting of potential complications of chlamydia infection including PID, EP, and TFI. The manuscript could benefit from a reframe as I had to read through a couple of times to piece together the information. Also somewhat related, better description of the current guidelines and health record systems in Netherlands would really help orient the non-Netherlands based reader.

Detailed suggestions/comments are listed below:

1. Abstract – the aims seem to be a bit misaligned with results/conclusion. Based on the results and discussion it seems that the purpose of the study is to use key informant interviews to understand ways to enhance a potential surveillance system for chlamydia complications. However, the objective seems to suggest a study that examines the impact of changes in chlamydia testing strategies. Please clarify.

2. Introduction –

a. The introduction would benefit from a more complete presentation of the debate around ‘test and treat’ for chlamydia. As it stands, the authors seem to suggest minimal benefit to chlamydia screening, whereas the literature suggests otherwise. In fact, the limitations in the literature are more specifically related to which screening strategies and intervals to use, which age to start and stop screening, and whether screening males is necessary.

b. Adding a brief overview of the current chlamydia testing guidelines and surveillance systems in the Netherlands would lend context to your study especially for readers outside of the Netherlands.

c. It seems the research question is whether there is a precedent or existing infrastructure that can be used to conduct surveillance for PID, EP, and TFI, the need for which may arise if widespread chlamydia screening was eliminated. If I’m correct, then perhaps inclusion of information on surveillance (in other settings) for PID/EP/TFI would be useful.

d. Paragraph on aims – as written this is unclear. Please rephrase.

3. Methods –

a. Sampling – could you clarify what you mean by stakeholders analysis. Do you mean a qualitative analysis?

b. Who conducted the interviews and were they all conducted by the same person?

4. Results/Discussion –

a. What is the added value of inclusion of the emergent themes related to diagnosis? Most of what is here is well established in the literature (e.g., difficulty in diagnosing PID).

b. Can you describe the standard practice/guidelines for registration of cases. What does registration mean? Presumably there currently is no surveillance system, but this is referring to medical record procedures and coding? Please clarify.

6. PLOS authors have the option to publish the peer review history of their article (what does this mean?). If published, this will include your full peer review and any attached files.

Reviewer #1: No

Reviewer #2: No

---

## [Author Response · Author response to Decision Letter 0]

19 Mar 2024

Dear editor, 

We have pleasure in submitting our revised manuscript entitled “Investigating possibilities for surveillance of long term chlamydia complications in the Netherlands: a qualitative study”. In this revised version, we have taken the reviewers’ comments into account. Below, we give a detailed response on the comments, and we hope to have satisfactorily addressed comments raised. We would be delighted if our revised version is now suitable for publication in PLOS ONE. 

Yours sincerely, 

Elke den Boogert, on behalf of all other co-authors. 

Response to reviewers

Reviewer #1

General comments:

Thank you for the opportunity to review this interesting manuscript on opportunities for surveillance of long term chlamydia complications in the Netherlands. The manuscript presents findings of a qualitative study among 15 healthcare professionals to explore their views on potential future routine surveillance of chlamydia related long-term complications, including pelvic inflammatory disease (PID), ectopic pregnancy (EP) and tubal factor infertility (TFI).

Given current debates on a paradigm shift from Chlamydia infection control to disease control, I believe that this is a timely topic of great public health relevance and general interest to PLOS ONE readers. 

Thank you for your time and interest in the manuscript. We are happy to read that the reviewer is positive about the subject and reviews it as an interesting and timely topic of great public health relevance.

However, I think that for the more general reader, it would be good if the authors could add further information to the introduction/discussion sections, including regarding limitations on the use of complications surveillance to compensate for reduced screening of asymptomatic infections. 

We agree and added a section to the discussion regarding the limitations of complication surveillance. Also see our response to comment #2c of reviewer 1. 

It is also important to acknowledge methodological limitations and ensure that relevant guidance on reporting of qualitative studies has been followed.

Based on the reviewers recommendation, we checked the manuscript according to the SRQR checklist and added the checklist to this letter. Changes we made according to the checklist can be found under comment #3a of reviewer 1. 

Specific comments:

1. Objectives in abstract and introduction:

I wonder whether the objectives could be stated a bit more clearly (possibly seeking support from a native speaker for this – well aware that I am not a native speaker, myself). I am not sure, for example, whether the word ‘opportunity’ is the best word to capture/specify what the authors mean – were you thinking of the word more in the sense of possibility/ options or in the sense of a favourable chance or circumstance and prospect for success? 

Thank you. We agree with the reviewer. We changed the word ‘opportunity’ to ‘possibilities’ in the objectives and throughout the manuscript (including the title). 

In the abstract, the objective ends with ‘aiming to evaluate the impact of changing chlamydia testing strategies on occurrence of complications.’ I think this refers to the routine surveillance of complications (i.e. complications surveillance could be used to evaluate the impact of potential future changes in testing strategies), but the quick reader might think that your qualitative study aims to evaluate to impact of changing testing strategies…

We removed the sentence in line 34 to avoid further confusion. 

2. Introduction/ Discussion:

a. You could consider briefly mentioning the prevalence and trends of Chlamydia complications globally and/or nationally (e.g. possibly He et al 2023 for the global burden of PID and EP, doi: 10.1186/s12889-023-16663-y)

Thank you, we updated the introduction regarding prevalence and incidence of the complications, line 57-58. 

“Globally, the age-standardized rates of PID prevalence was 53.19 per 100,000 population in 2019 and for EP it was 342.22 per 100,000 population (3).”

b. The general reader might be interested to know whether any other countries have already changed testing strategies and/or have already some form of Chlamydia complications surveillance in place and what their experiences is with this?

We updated the introduction in line 80-84 saying: “Some countries have already implemented small changes to their testing guidelines, e.g. focussing more on reducing harms from untreated chlamydia infection rather than onward transmission (13). Australia replaced its focus from testing uptake to strengthening management of diagnosed infections, which involves improving retesting, partner management and PID diagnosis (14).”

We added a statement to the introduction in line 70-71: “To our knowledge, no country has implemented routine surveillance of chlamydia related complications.”

c. You could consider discussing surveillance of Chlamydia complications with regards to certain limitations, including relating to the correlation of Chlamydia infections with PID and with subsequent EP and/or TFI. For example, trends could be discordant due to changes in the health care system other than STI testing, e.g. regarding delayed or inadequate treatment of Chlamydia infection and/or PID. (For changes in access to hospital care, see for example the paper by Chen et al 2005 doi: 10.1136/sti.2004.012807 ). In the case of TFI, possibly changing pregnancy rates would also need to be considered when analysing data.

Thank you. This was not a theme raised by the interviewees, however important to address. We added a section to the discussion in line 333-340. 

“When considering complication surveillance, an important aspect should be taken into account. Possible observed changes in PID, EP and/or TFI trends cannot be directly linked to changes in incidence of chlamydia infections, as there could be many other explanations for the observed trends. Especially since other infections can also lead to PID, EP and TFI, such as gonorrhoea (18). Other explanations could include changes in access to health care or changes in treatment of chlamydia infections (17). Therefore one should be careful in interpreting trends of PID, EP and TFI when complication surveillance is in place. ”

d. It would also be good if the more general reader could have an idea on how likely it is that it can be differentiated that PID, EP and/or TFI are due to Chlamydia, and not due to other infections, such as Gonorrhoea (also keeping gonorrhoea-related antimicrobial drug resistance in mind), or bacterial vaginosis-associated bacteria, or other causes, such as endometrioses (keeping changing trends in endometriosis rates in mind)? Having said all this, I believe that PID, EP and TFI changes are worth monitoring, regardless of its cause, but issues around cost-effectiveness might need to be discussed.

Thank you for this suggestion we included in the introduction in line 58-60: “A study in the United Kingdom estimates that every 1000 chlamydia infections in women aged 16-44 years, on average, results in 171 extra episodes of PID, 2 more EP and 5.1 more women with TFI at age 44 years (4).” 

e. Line 264-265 reads “Based on what respondents told us, we feel that gynaecologists, more than GPs, actually diagnose the complications, and properly register them in their hospital databases”. International readers might not be sure why gynaecologists would be able to register complications in ‘their hospital database’, as they might not know whether in the Netherlands gynaecologists are usually based at hospitals (including for outpatient care) or in (private or public) clinics/ practices separate from hospitals. It would therefore be good to briefly provide in the introduction or discussion some background information on the Dutch health care system, also in terms of access to free health care.

We recognize the lack of information on the Dutch context and added a dedicated section in the methods section, line 101-108. 

“Dutch health care context

The Dutch health care system is based on universal access to care, including compulsory medical insurance for all. All residents of the Netherlands can access care through a basic health insurance package. On top of the mandatory nominal premium, individuals are required to pay a deductible, an amount that insured individuals must pay out of pocket each year before their health insurer begins to reimburse expenses. Some care (e.g. at the General Practitioner) is excluded from co-payment. The General Practitioner (GP) acts as a gatekeeper and can make referrals to medical specialists. Medical specialists (including gynaecologists) are linked to public hospitals (17).”

f. Generally, I think the manuscript is well written, I have only spotted several typos/ grammar/spelling errors, and it is necessary to cross-check the whole document, e.g. in line 135, 143,147 and 256 etc need to say “patient’s” or “patients’” instead of “patients”, line 113 reads “in-dept interviews” instead of “in-depth interviews”, line 188 should say “tubal pathology” instead of “tuba pathology”, and in line 165 please clarify what is meant by “red.” In “red. perform an STI test…”. In line 83, please indicate the meaning of the acronym SHC at first use, even if most readers will guess what is meant. In line 240 it is not clear what ‘NIVEL’ means.

Thank you for carefully reading the manuscript. We updated the errors throughout the manuscript, checked the manuscript for more, and corrected accordingly. The Red. in the quote was added as a clarification to what the ‘that’ was referring to. ‘Red.’ can be used to indicate that the text in brackets was added by the authors to clarify the statement. We deleted the Red. as we think it is clear without it. 

We added an explanation to what NIVEL means in the discussion in line 295-296. “a sentinel surveillance system, called Nivel (the Netherlands Institute for Health Services Research) sentinel stations”. 

3. Methods:

a. Following reporting guidance: Guidance by different journals varies, but I think it would be good if you could indicate at the beginning of the methods section that you have used relevant international guidelines/checklists for the reporting of qualitative research. Therefore, if the editor agrees, please upload with your rebuttal letter a SRQR checklist(http://journals.lww.com/academicmedicine/fulltext/2014/09000/Standards_for_Reporting_Qualitative_Research___A.21.aspx) or the COREQ checklist (https://academic.oup.com/intqhc/article/19/6/349/1791966/Consolidated-criteria-for-reporting-qualitative), indicating the page/line numbers of your manuscript where the relevant information can be found, updating the manuscript as needed to ensure all reporting requirements are met.

For example, I noted that there is insufficient information on your qualitative approach/ research paradigm and researcher characteristics and reflexivity.

Thank you. We checked our manuscript according to the SRQR checklist and added the checklist to this letter (A Word file named SRQR_Checklist). We updated the manuscript, specifically on the information gaps the reviewer mentioned, by adding “The research team consists of three epidemiologists working in infectious diseases and STI research. The interviewer had no prior interaction with the respondents regarding the themes described, however some respondents were part of the network of the research team, the interviewer and other respondents” and “An interpretive qualitative research approach was used for data collection by developing semi-structured interview guide for individual face-to-face or online interviews.” to the methods section in line 141-145. 

b. Sample size and saturation: Line 85 indicates that the sample size was determined by the principle of saturation and that where saturation was not reached or was difficult, medical doctors working under supervision of a gynaecologist were recruited. Overall, I think that the sample sizes for the different specialities in different regions of the country are rather low and it seems unlikely that saturation (however defined) has been reached for all relevant concepts (although it might have been met for some of the most important themes, but this would need to be more clearly stated). Generally, it seemed that the reported results are quite descriptive and do not go much into depths. Nevertheless, I think you have retrieved important information from this research, it may only need further information and better acknowledgement of its limitations. For example, you found that one of the six GPs said they sometimes diagnose and treat PID without referral to a gynaecologist, which would mean that the diagnosis would not be captured by gynaecologists/ in a hospital database. It is unclear whether in a bigger sample, more GPs would have reported similar behaviour.

We acknowledge this limitation in the discussion and included in the manuscript (line 284-292):

“Based on the interview with the ER doctor, the interviews with other stakeholders, and informal communication, we decided not to include more ER doctors. We concluded that they were not relevant stakeholder in the diagnosis and registration of complications in the Netherlands. Saturation for SHC doctors was reached after two interviews, a third interview was conducted to confirm saturation. Furthermore, we believed that saturation for GPs and gynaecologists was reached for the main themes. For example, professionals were clear that diagnosis of complications would not change if chlamydia testing strategies would change(main theme 1). For other topics, such as whether or not all GPs would refer a suspected PID to the gynaecologist, a larger sample size could yield a wider range of views than reported in this article. ”

The concept of ‘data saturation’ is well-debated in qualitative methodology literature and it would therefore be good if you could define this term more clearly and your decisions around it in your paper. Did you have an initial sample size you were aiming for? How did you decide (before and/or during data collection) your definition of data saturation to conclude recruitment, and how did you measure this? How could you be sure there were no more relevant themes to come, or that further interviews could not contribute to refining of the themes? Please see for example the references below regarding the current debate on ‘saturation’ and sample sizes in qualitative research:

Saunders B, et al. Saturation in qualitative research: exploring its conceptualization and operationalization. Qual Quant. 2018;52(4):1893-1907. Epub 2017 Sep 14. DOI: 10.1007/s11135-017-0574-8

Tight, M. (2023). Saturation: An Overworked and Misunderstood Concept? Qualitative Inquiry, 0(0). https://doi.org/10.1177/10778004231183948

Malterud K, et al: Sample Size in Qualitative Interview Studies: Guided by Information Power. Qual Health Res 2016, 26(13):1753-1760. DOI: 10.1177/1049732315617444

Thank you for the interesting read on the data saturation. We changed the statement on saturation in the methods section in line 116-117 to the following “The sample size was determined by the principle of saturation for the main themes of the interview guide (see S1 for interview guide).”

The main themes were added to the methods section (line 127-136), for more information see our response to reviewer 2 comment 4a. 

Reviewer #2

General comments

Thank you for the opportunity to review this manuscript which uses qualitative methods to examine clinician practices and behaviors related to diagnosis and reporting of potential complications of chlamydia infection including PID, EP, and TFI. The manuscript could benefit from a reframe as I had to read through a couple of times to piece together the information. 

Thank you for carefully reading our manuscript. We made a lot of changes to the manuscript according to the helpful comments of both reviewers. Thanks to this, we feel the manuscript reads much better now. 

Also somewhat related, better description of the current guidelines and health record systems in Netherlands would really help orient the non-Netherlands based reader.

We added a dedicated section to the methods about the Dutch health care context, regarding 

---

## [Decision Letter · Decision Letter 1]

8 Apr 2024

PONE-D-23-23499R1Investigating possibilities for surveillance of long term chlamydia complications in the Netherlands: a qualitative studyPLOS ONE

Dear Dr. den Boogert,

Thank you for submitting your manuscript to PLOS ONE. After careful consideration, we feel that it has merit but does not fully meet PLOS ONE’s publication criteria as it currently stands. Therefore, we invite you to submit a revised version of the manuscript that addresses the points raised during the review process.

Please address Reviewer 1's comments regarding additional description of your methods for qualitative analysis and sample size, and inclusion of the SRQR checklist. Please also complete a final check of the manuscript for grammar and spelling prior to resubmission.==============================

We look forward to receiving your revised manuscript.

Kind regards,

Jesse L Clark, MD, MSc

Academic Editor

PLOS ONE

Journal Requirements:

Reviewers' comments:

Reviewer's Responses to Questions

**Comments to the Author**

1. If the authors have adequately addressed your comments raised in a previous round of review and you feel that this manuscript is now acceptable for publication, you may indicate that here to bypass the “Comments to the Author” section, enter your conflict of interest statement in the “Confidential to Editor” section, and submit your "Accept" recommendation.

Reviewer #1: (No Response)

Reviewer #2: All comments have been addressed

2. Is the manuscript technically sound, and do the data support the conclusions?

Reviewer #1: (No Response)

Reviewer #2: Yes

3. Has the statistical analysis been performed appropriately and rigorously? 

Reviewer #1: (No Response)

Reviewer #2: N/A

4. Have the authors made all data underlying the findings in their manuscript fully available?

Reviewer #1: (No Response)

Reviewer #2: (No Response)

5. Is the manuscript presented in an intelligible fashion and written in standard English?

Reviewer #1: (No Response)

Reviewer #2: Yes

6. Review Comments to the Author

Reviewer #1: Thank you for responding to the reviewer questions. I think the manuscript has significantly improved and may be published after the points below have been addressed:

Methods section

1. Line 121 – Thank you for adding the below in response to my previous comment regarding which qualitative approach/research paradigm you have used: “An interpretive qualitative research approach was used for data collection by developing a semi-structured interview guide…”. Would you mind elaborating on this approach and providing a reference for this? Also, you have added this to the ‘data collection’ section, but since a research paradigm usually does not only relate to data collection, but also to data analysis and interpretation, it would be good if you could elaborate how your overall research paradigm influenced also data analysis and interpretation.

2. The authors have indicated that they have checked the “manuscript according to the SRQR checklist and added the checklist to this letter (A Word file named SRQR Checklist)” – However, the checklist has not been included in the pdf file and I have not been able to find the SRQR checklist elsewhere in the system for some reason…

Discussion

3. Line 287 – First, thank you for partly addressing my previous question regarding sample size and saturation by making changes to the methods section and the discussion. This is now clearer, apart from line 287, where you say “Saturation for SHC doctors was reached after two interviews, a third interview was conducted to confirm saturation.” Instinctively, the reader might think that after only two interviews, there might still be a high chance of missing important themes or perspectives – although in a highly homogenous sample, such as SHC doctors, this might be less of an issue, if referring only to ‘main themes’, as you had done for GPs; there might also be sampling bias for the first few participants, especially if a ‘network/ snowball sampling’ approach has been used. I would therefore be careful with such statements, or justify them better with reference to the literature (including the literature I had referred to during my previous review).

Entire manuscript

4. I have spotted a number of grammar mistakes – for example:

- Abstract, line 33 – Instead of “the possibility to introducing” it should say “the possibility of introducing”

- Introduction, line 91 – Instead of “monitor trends in chlamydia.” it should say something like “monitor trends in chlamydia infections.”

- Line 96 – Instead of “More specific,, we aimed to understand...” it should read “More c, we aimed to understand…”

After the revision, the entire manuscript will therefore need to be cross-checked for typos/ grammar mistakes.

Reviewer #2: The reviewers have addressed all comments raised as part of the original review. I have just a few minor comments:

Line 81 - there is a word missing in this sentence

Line 84 - missing punctuation

Line 217-219 - this sentence is unclear; CAT should be defined on first use

7. PLOS authors have the option to publish the peer review history of their article (what does this mean?). If published, this will include your full peer review and any attached files.

Reviewer #1: No

Reviewer #2: No

---

## [Author Response · Author response to Decision Letter 1]

23 May 2024

Thank you for giving us the opportunity to resubmit our manuscript entitled “Investigating possibilities or surveillance of long term chlamydia complications in the Netherlands: a qualitative study”. Please find our response to the editorial and reviewer comments below. 

On behalf of all authors,

Elke den Boogert

Response to editorial comments:

Thank you for pointing this out. We double checked our reference list and we saw that reference 19 is in the reference list but not in the text. Reference 18 in line 349 should have been reference 19. We now changed it to reference 23 in line 349, as we also added four references earlier in the manuscript (line, 106-107, 109, 154 and 299). 

Response to reviewer comments

Reviewer #1: Thank you for responding to the reviewer questions. I think the manuscript has significantly improved and may be published after the points below have been addressed:

Methods section

1. Line 121 – Thank you for adding the below in response to my previous comment regarding which qualitative approach/research paradigm you have used: “An interpretive qualitative research approach was used for data collection by developing a semi-structured interview guide…”. Would you mind elaborating on this approach and providing a reference for this? 

Thank you. We updated the manuscript by including a section on study design in the

methods in line 105-111: 

“Design

A qualitative interpretative phenomenological method was used (18). Phenomenology seeks to understand the individual interpretation and view on a phenomenon (18). In our study we were interested in individual interpretation of guidelines and registration. Diagnosis of complications is set in guidelines, however interpretation seems difficult (19). To better understand the experiences of professionals in applying these guidelines and related registrations in their daily practice, we conducted phenomenological interviews.”

Also, we added the approach to line 113 ‘This interpretive phenomenological qualitative study included …’

Also, you have added this to the ‘data collection’ section, but since a research paradigm usually does not only relate to data collection, but also to data analysis and interpretation, it would be good if you could elaborate how your overall research paradigm influenced also data analysis and interpretation.

Thank you. We added more information about the design to the methods, see our answer above. Also, we added to line 152-154 ‘To identify individual experience with interpreting guidelines and registrations in our qualitative dataset we conducted an interpretive thematic analysis using MAXQDA Plus 2022 (Release 22.0.0) for data management (20).’

2. The authors have indicated that they have checked the “manuscript according to the SRQR checklist and added the checklist to this letter (A Word file named SRQR Checklist)” – However, the checklist has not been included in the pdf file and I have not been able to find the SRQR checklist elsewhere in the system for some reason…

We added the SRQR checklist as ‘Other’ in the submission system and was available to access at the end of the pdf file. We updated the checklist according the last comments from the reviewer and also added it below this rebuttal letter as well as ‘Other’ material in the system. 

Discussion

3. Line 287 – First, thank you for partly addressing my previous question regarding sample size and saturation by making changes to the methods section and the discussion. This is now clearer, apart from line 287, where you say “Saturation for SHC doctors was reached after two interviews, a third interview was conducted to confirm saturation.” Instinctively, the reader might think that after only two interviews, there might still be a high chance of missing important themes or perspectives – although in a highly homogenous sample, such as SHC doctors, this might be less of an issue, if referring only to ‘main themes’, as you had done for GPs; there might also be sampling bias for the first few participants, especially if a ‘network/ snowball sampling’ approach has been used. I would therefore be careful with such statements, or justify them better with reference to the literature (including the literature I had referred to during my previous review).

Thank you for this further explanation and comment. We changed the section in line 294-299. The section now reads: “Saturation for the main themes was reached after two interviews with SHC doctors, mainly since they explained that they hardly ever diagnose chlamydia related complications. A third interview was conducted to confirm their responses regarding the main themes. However, since network and snowball sampling could result in a more limited specificity, and thus requiring more participants to reach saturation, it might still be possible that other views among SHC doctors exist (21).

Entire manuscript

4. I have spotted a number of grammar mistakes – for example:

- Abstract, line 33 – Instead of “the possibility to introducing” it should say “the possibility of introducing”

- Introduction, line 91 – Instead of “monitor trends in chlamydia.” it should say something like “monitor trends in chlamydia infections.”

- Line 96 – Instead of “More specific,, we aimed to understand...” it should read “More c, we aimed to understand…”

After the revision, the entire manuscript will therefore need to be cross-checked for typos/ grammar mistakes.

Thank you. We changed the abstract and the introduction as suggested. We checked the entire manuscript and made additional small changes throughout the manuscript. 

It is unclear to what mistake the reviewer is referring in line 96. We don’t see a grammar mistake. 

Reviewer #2: The reviewers have addressed all comments raised as part of the original review. I have just a few minor comments:

Line 81 - there is a word missing in this sentence

Line 84 - missing punctuation

Line 217-219 - this sentence is unclear; CAT should be defined on first use

Thank you for your comments. We added the word ‘for’ in line 81, the sentence now reads: “In the Netherlands, guidelines recommend chlamydia testing for persons experiencing symptoms, their contacts, and risk groups regardless of symptoms such as persons who had unprotected sex (15).”

We added the punctuation in line 85. 

CAT is defined in line 216, when it is used for the first time. For clarity, we also added the definition to line 231.

---

## [Editor Report · Decision Letter 2]

28 May 2024

Investigating possibilities for surveillance of long term chlamydia complications in the Netherlands: a qualitative study

PONE-D-23-23499R2

Dear Dr. den Boogert,

We’re pleased to inform you that your manuscript has been judged scientifically suitable for publication and will be formally accepted for publication once it meets all outstanding technical requirements.

Kind regards,

Jesse L Clark, MD, MSc

Academic Editor

PLOS ONE

---

## [Editor Report · Acceptance letter]

31 May 2024

PONE-D-23-23499R2 

PLOS ONE

Dear Dr. den Boogert, 

I'm pleased to inform you that your manuscript has been deemed suitable for publication in PLOS ONE. Congratulations! Your manuscript is now being handed over to our production team.

Kind regards, 

on behalf of

Dr. Jesse L Clark 

Academic Editor

PLOS ONE